# A New Breakpoint to Classify 3D Voxels in MRI: A Space Transform Strategy with 3t2FTS-v2 and Its Application for ResNet50-Based Categorization of Brain Tumors

**DOI:** 10.3390/bioengineering10060629

**Published:** 2023-05-23

**Authors:** Hasan Koyuncu, Mücahid Barstuğan

**Affiliations:** Electrical & Electronics Engineering Department, Faculty of Engineering and Natural Sciences, Konya Technical University, Konya 42250, Türkiye; mbarstugan@ktun.edu.tr

**Keywords:** brain, convolutional neural network, dimensional, feature transform, glioma grading, image classification, transfer learning, tumor

## Abstract

Three-dimensional (3D) image analyses are frequently applied to perform classification tasks. Herein, 3D-based machine learning systems are generally used/generated by examining two designs: a 3D-based deep learning model or a 3D-based task-specific framework. However, except for a new approach named 3t2FTS, a promising feature transform operating from 3D to two-dimensional (2D) space has not been efficiently investigated for classification applications in 3D magnetic resonance imaging (3D MRI). In other words, a state-of-the-art feature transform strategy is not available that achieves high accuracy and provides the adaptation of 2D-based deep learning models for 3D MRI-based classification. With this aim, this paper presents a new version of the 3t2FTS approach (3t2FTS-v2) to apply a transfer learning model for tumor categorization of 3D MRI data. For performance evaluation, the BraTS 2017/2018 dataset is handled that involves high-grade glioma (HGG) and low-grade glioma (LGG) samples in four different sequences/phases. 3t2FTS-v2 is proposed to effectively transform the features from 3D to 2D space by using two textural features: first-order statistics (FOS) and gray level run length matrix (GLRLM). In 3t2FTS-v2, normalization analyses are assessed to be different from 3t2FTS to accurately transform the space information apart from the usage of GLRLM features. The ResNet50 architecture is preferred to fulfill the HGG/LGG classification due to its remarkable performance in tumor grading. As a result, for the classification of 3D data, the proposed model achieves a 99.64% accuracy by guiding the literature about the importance of 3t2FTS-v2 that can be utilized not only for tumor grading but also for whole brain tissue-based disease classification.

## 1. Introduction

Three-dimensional (3D) imaging is inevitably used in medical areas including brain tumor analyses. Magnetic resonance imaging (MRI) is the most frequently used imaging modality mainly owing to its ability to differentiate among different soft tissues. Moreover, this feature makes the usage of MRI inevitable for brain tumor-based segmentation or classification systems. Regarding this, in the literature, non-invasive-based studies generally focus on the segmentation process of MRI data for various kinds of brain tumors [1,2,3].

The classification of brain tumors poses a significant challenge for designing a computer-aided diagnosis (CAD) system that is fully automated. For this purpose, it is expected that a classification model can be adapted for using 3D tumor-based data since it should be applicable to the 3D output of the segmentation part in the fully automated CAD. However, in the literature, many studies examine two-dimensional (2D) data, which require the choice of a slice or 2D image. In other words, an expert should examine the tumorous/non-tumorous regions in 3D data [4,5,6,7,8,9,10]. At this point, this process yields the system to be a semi-automated CAD, which needs an external intervention to start the evaluation process [4,5,6,7,8,9,10]. In the literature, MRI-based classification is usually performed using 2D information for multiclass tumor types [4,5,6,7,8,9,10], normal/abnormal labels [11,12], tumor/non-tumor categories [13], and a combination of these applications [13].

The most frequent brain tumors (glioma) are generally divided into two important categories that are high-grade glioma (HGG) and low-grade glioma (LGG) [14,15]. Herein, LGG-type tumors are the second important category, although they are not a priority, but they should be identified. However, HGG-type tumors constitute the first priority to be detected at an early stage considering the survival of the patients. Despite the importance of this issue, very few studies have handled just the high-grade vs. low-grade discrimination of gliomas using 2D or 3D information in the literature [16]. 

In the literature studies about HGG/LGG discrimination, deep learning-based models or task-specific frameworks were generally utilized/designed to assign the tumors to HGG and LGG classes. Koyuncu et al. [16] proposed an efficient framework classifying 210 HGG and 75 LGG samples defined using 3D MRI data. For this purpose, first-order statistics (FOS) were evaluated to extract the features from the 3D tumor voxel. Herein, three-phase (T1, T2, FLAIR) information was found to be the best combination in trials with the BraTS 2017/2018 dataset. Feature selection was fulfilled using Wilcoxon ranking. An optimized classifier named GM-CPSO-NN was considered to perform the categorization. As a result, the task-specific framework achieved a 90.18% accuracy on HGG/LGG distinction. Mzoughi et al. [17] suggested a 3D deep convolutional neural network (3D Deep CNN) including data augmentation to discriminate the HGG/LGG tumors using the BraTS 2017/2018 dataset. The T1c sequence was used as the input of the system. Consequently, the deep learning-based model obtained a 96.49% accuracy for the categorization of brain tumors. Tripathi and Bag [18] combined four residual networks (ResNet18, ResNet50, ResNet101, and ResNet152) by using a novel Dempster–Shafer theory (DST). In [18], the cancer imaging archive (TCIA) library was evaluated involving both HGG and LGG samples. In addition, T2 phase information was considered to form the input data for ResNets fusion. In experiments, a 95.87% accuracy was observed with the deep learning-based model for the classification of 2D images that comprised only tumor tissue. Montaha et al. [19] generated a deep learning-based model in which 3D CNN and long short-term memory (LSTM) with a TimeDistributed (TD) function were operated. In experiments, the BraTS 2020 dataset was utilized as the test dataset incorporating 234 HGG and 74 LGG samples in all phases (T1, T2, T1ce, FLAIR). The 3D-based classification model (TD-CNN-LSTM) recorded a 98.90% accuracy for HGG/LGG classification using 3D MRI data. Jeong et al. [20] presented a model that determined a multimodal fusion network using adversarial learning. The BraTS 2017/2018 dataset was evaluated with all phase information to test the system performance. In [20], the model was defined as a 2.5D-based classification system since it only focused on a few slices of the tumor. According to the results, the proposed model acquired a 90.91% accuracy for HGG/LGG classification. Bhatele and Bhadauria [21] generated a task-specific model by using five feature extraction approaches that are the discrete wavelet transform (DWT), gradient gray level co-occurrence matrix (GGLCM), local binary patterns (LBP), gray level run length matrix (GLRLM), and morphological features. Principal component analysis (PCA) was utilized to decrease the feature number. All phase information was fed to the input of an ensemble classifier. Regarding the results, the 2D-based classification model achieved 100% and 99.52% accuracy scores for the BraTS 2013 and BraTS 2015 datasets, respectively. Demir et al. [22] designed a new deep learning-based model (3ACL) that combines the 3D Attention module, CNN, and LSTM in the 3D-based categorization of HGG and LGG labels. In [22], all phase information was evaluated using 3D MRI data, and the classifier unit was formed on the basis of support vector machine (SVM) algorithms followed by a weighted majority vote. Concerning the experiments, 3D-based deep learning architecture yielded 98.90% and 99.29% classification accuracies, respectively, for BraTS 2015 and BraTS 2018 datasets. Hajmohamad and Koyuncu [23] recommended a new feature transform strategy (3t2FTS) to convert the information of 3D space into 2D space. In other words, 3D tumor voxels were transformed into the 2D-identity (2D-ID) images for every tumor defined using 3D MRI. For this purpose, all phase information and FOS features were evaluated to form the 2D-ID images. In this way, eight traditional transfer learning models (DenseNet201, Inception-ResNetV2, InceptionV3, ResNet50, ResNet101, SqueezeNet, VGG19, and Xception) were applicable to 2D data and were compared to reveal the most coherent one with the 3t2FTS approach. In tests with the BraTS 2017/2018 dataset, ResNet50 architecture provided an 80% classification accuracy for HGG/LGG discrimination.

As seen in the literature studies, deep learning-based models are effectively utilized to detect the type of brain tumors [23,24]. On the other hand, HGG/LGG discrimination is barely handled in the literature by means of 2D-, 2.5D-, and 3D-based classification tasks [16,17,18,19,20,21,22,23]. To the best of our knowledge, the classification part of a fully automated CAD can only be obtained if the information of 3D tumors is directly examined with efficient models. At this point, it is revealed that a few studies determine the 3D-based HGG/LGG classification with deep learning architectures [17,19,22,23]. However, according to our examinations, there is no feature transform strategy (except 3t2FTS) to applicate the transfer learning models for the classification of tumors in a 3D-defined space.

Concerning the examinations, there exist the following inferences:ResNet50 or ResNet-based architectures are frequently applied to perform brain tumor classification using MRI data [6,18,23].3D-based HGG/LGG categorization arises as a remarkable issue to design the classification part of a fully automated CAD.Except for 3t2FTS, a novel feature transform strategy is not available in the literature to applicate 2D-based transfer learning models and to find the 2D-ID image of a 3D voxel.

Concerning the inferences, the motivation of this paper and its literature contributions are revealed as:The design of a state-of-the-art feature transform strategy (3t2FTS-v2) to transform the 3D space information to the 2D space.The applicable transform strategy to be considered not only for 3D-defined tumors but also for the whole brain defined in 3D MRI (disease classification).A case study using FOS, GLRLM, and normalization analyses to discover 2D-ID images of 3D voxels.A comprehensive research obtaining promising results on 3D-based HGG/LGG categorization.An extensive study about ResNet50 and its hyperparameter adjustments on brain tumor classification in 3D MRI.

The paper is organized as follows. Section 2 briefly explains the FOS and GLRLM features, comprehensively determines the formation of the 3t2FTS-v2 approach, summarizes the ResNet50 architecture, and clarifies the dataset information with its handicaps. Section 3 describes the experimental analyses and interpretations in detail. Section 4 presents the discussions and literature comparison. Section 5 concludes the paper.

## 2. Materials and Methods

### 2.1. First-Order Statistics

First-order statistics (FOS) are effectively utilized in the literature for texture-based 2D and 3D analyses. Herein, FOS features are produced considering the intensity- or (mostly) histogram-based evaluations of an image. On intensity-based examinations, the features are directly obtained using the FOS along the image. On histogram-based examinations, the histogram of 2D or 3D data is generated, and FOS features are extracted along the histogram [23,25].

Let *y*, *x*, *f*(*x*,*y*), *G*, and *i* symbolize the vertical plane, horizontal plane, the function specifying a 2D image using the coordinate of (*x*,*y*), the total intensity number in image, and the output of *f*(*x*,*y*) or discrete intensity having a numerical value within the intensity levels of [0, *G* − 1], respectively. Then, the histogram analyses are performed considering the repetition number of intensity levels along the whole image. If the histogram-based analysis is intended to be used for a 3D voxel, then the size of an image should be defined in 3D space. If *N*, *M*, and *L* represent the length, width, and slice number of the image, respectively, then the total number of voxels can be obtained using the volume of interest (*VOI*) information with *M* × *N* × *L* multiplication. Regarding this, the histogram of the *i*th intensity value described as *h*(*i*) can be found as in (1), whilst the Kronecker Delta function symbolized as *δ*(*i*,*j*) can be calculated using Equation (2). In addition, the probability density function (PDF) of *i* is evaluated using Equation (3) by dividing *h*(*i*) by the *M* × *N* × *L* multiplication (the total voxel number belonging to VOI) [23,25].
(1)$$h(i)=\sum_{x=0}^{N-1}\sum_{y=0}^{M-1}\delta (f(x,y),i)$$
(2)$$\delta (j,i)=\left\{\begin{array}{l} 1,j=i\\ 0,j\neq i \end{array}\right.$$
(3)$$p(i)=\frac{h(i)}{1\times L\times M\times N},\;i=0,1,2,3\ldots ,G-1$$

By considering the PDF information apart from the aforementioned phenomena, FOS features of a VOI are calculated as the mean, standard deviation, skewness, kurtosis, energy, and entropy that are identified in Equations (4)–(9), respectively [23,25].
(4)$$\mu =\sum_{i=0}^{G-1}ip(i)$$
(5)$$\sigma =\sqrt{\sum_{i=0}^{G-1}{\left(i-\mu \right)}^{2}p(i)}$$
(6)$$\mu_{3}=\sigma^{-3}\sum_{i=0}^{G-1}{\left(i-\mu \right)}^{3}p(i)$$
(7)$$\mu_{4}=\sigma^{-4}\sum_{i=0}^{G-1}{\left(i-\mu \right)}^{4}p(i)-3$$
(8)$$Energy=\sum_{i=0}^{G-1}{\left[p(i)\right]}^{2}$$
(9)$$Entropy=-\sum_{i=0}^{G-1}p(i)\log_{2}\left[p(i)\right]$$

### 2.2. Gray Level Run Length Matrix

The gray level run length matrix (GLRLM) calculates the number of homogeneous runs along each gray level *i*. Let *G*, *R*, and *N* symbolize the gray level number, the longest run, and the pixel number, respectively. In this way, GLRLM can be seen as a 2D matrix at the size of *G* × *R*. Herein, the elements of each *p*(*i*,*j*|*θ*) arise as the occurrence value of runs with the *θ*th direction, *i*th gray level, and *j*th run length, respectively. Seven general GLRLM features are defined as the long runs emphasis (*LRE*), short runs emphasis (*SRE*), run length non-uniformity (*RLN*), gray level non-uniformity (*GLN*), low gray level runs emphasis (*LGRE*), high gray level runs emphasis (*HGRE*), and run percentage (RP). These features are respectively obtained using Equations (10)–(16) [25,26]. In our paper, six of the seven aforementioned features (except *HGRE*) are utilized, since these provide an information change among 2D-ID images for every 3D-defined tumor.
(10)$$LRE=\sum_{i=1}^{G}\sum_{j=1}^{R}j^{2}p(i,j|\theta )/\sum_{i=1}^{G}\sum_{j=1}^{R}p(i,j|\theta )$$
(11)$$SRE=\sum_{i=1}^{G}\sum_{j=1}^{R}\frac{p(i,j|\theta )}{j^{2}}/\sum_{i=1}^{G}\sum_{j=1}^{R}p(i,j|\theta )$$
(12)$$RLN=\sum_{j=1}^{R}{\left(\sum_{i=1}^{G}p(i,j|\theta )\right)}^{2}/\sum_{i=1}^{G}\sum_{j=1}^{R}p(i,j|\theta )$$
(13)$$GLN=\sum_{i=1}^{G}{\left(\sum_{j=1}^{R}p(i,j|\theta )\right)}^{2}/\sum_{i=1}^{G}\sum_{j=1}^{R}p(i,j|\theta )$$
(14)$$LGRE=\sum_{i=1}^{G}\sum_{j=1}^{R}\frac{p(i,j|\theta )}{i^{2}}/\sum_{i=1}^{G}\sum_{j=1}^{R}p(i,j|\theta )$$
(15)$$HGRE=\sum_{i=1}^{G}\sum_{j=1}^{R}i^{2}p(i,j|\theta )/\sum_{i=1}^{G}\sum_{j=1}^{R}p(i,j|\theta )$$
(16)$$RP=\frac{1}{N}\sum_{i=1}^{G}\sum_{j=1}^{R}p(i,j|\theta )$$

### 2.3. Design of 3t2FTS-v2

The 3D to 2D feature transform strategy (3t2FTS) has been suggested by Hajmohamad and Koyuncu [23] to convert the features in 3D space into the 2D plane. In the 3t2FTS or 3t2FTS-v1 approach, FOS features are evaluated to produce the 2D-ID images acting as an identity that belongs to 3D voxels [23]. However, only FOS features are considered in the transformation. In addition, any other feature extraction methods and normalization approaches are not examined that can generate robust 2D-ID images by providing discriminative information.

In this paper, we propose the second version of 3t2FTS (3t2FTS-v2) in which FOS and GLRLM are handled in addition to the non-normalization-based and normalization-based analyses. Thereby, discriminative 2D-ID images that are more meaningful are generated using 3t2FTS-v2 to transform the 3D MRI information into 2D space. Herein, it can be seen that the motivation and design of the method can be applied to both a 3D-defined tumor and 3D brain tissue. Moreover, both versions can be implemented for the evaluation of not only tumors but also brain diseases such as Alzheimer, etc. defined in 3D images. In this way, general machine learning methods (deep learning, transfer learning, etc.) that are designed on the basis of 2D evaluation can be activated using both versions. Figure 1 shows the design of the 3t2FTS-v2 approach.

As seen in Figure 1,

In item (1), the tumor area is obtained by multiplying the tumor mask of BraTS 2017/2018 with the 3D MRI voxel as in 3t2FTS. However, 3t2FTS-v2 operates an additional part providing data cleaning in null slices that all pixels own some non-zero (close to zero) values. Herein, this situation can change the meaningful information in the 2D-ID image. Concerning this, the non-zero null slices are converted to the matrixes including zero values by considering the standard deviation along the image.In item (2a), six FOS features (mean, standard deviation, skewness, kurtosis, energy, and entropy) are generated for each slice. Regarding this, meaningful information is produced at the size of 6 × 155.In item (2b), six GLRLM features (SRE, LRE, GLN, RLN, RP, and LGRE) are evaluated for every slice. Concerning this, distinctive information is generated at the size of 6 × 155.In items (2a) and (2b), it should be reminded that location information is processed in addition to the intensity-based, size-based, and shape-based features.In item (3), the outcomes of items (2a) and (2b) are combined to form the information at the size of 12 × 155.In item (4), the previous items (1, 2a, 2b, and 3) are respectively applied for every MRI sequence. Consequently, four information matrixes belonging to all MRI phases are independently obtained at the size of 12 × 155 for one tumor, individually.In item (5), *z-score* normalization is fulfilled for every row in the data, independently. This process yields the normalization of every feature in itself and the feature transform is performed more robustly. Herein, item (5) is performed separately for all the 12 × 155 information in all phases.In item (6), the normalized information matrixes at the size of 12 × 155 are combined to discover the 2D-ID image of a 3D tumor.

As seen at the end of Figure 1, three examinations are performed to design the 3t2FTS-v2. These examinations comprise the options of non-normalization, *minmax* normalization, and *z-score* normalization, all of which diversely generate different 2D-ID images. In experiments, *z-score* normalization comes to the forefront in terms of obtaining the best scores and being the most coherent one to transform the feature space. Moreover, design items of the algorithm are considered by evaluating the experiments and results of our study.

### 2.4. ResNet50 Architecture

For the design of an improved CNN, the performance of the architecture can be upgraded if the number of layers is increased in comparison with the original version. However, the deeper networks can cause a degradation of accuracy among training iterations (vanishing gradient problem). Herein, residual network (ResNet)-based architectures are proposed to prevent this handicap from occurring from the nature of the design [23,27]. 

Figure 2 presents the general design of ResNet50 architecture. As seen in Figure 2, ResNet50 is proposed on the basis of five convolution modules involving convolution layers of different sizes. In addition, it utilizes the maximum and average pooling approaches, the fully connected neural network (NN) layer, and a *softmax* function [23,27].

ResNets or ResNet-inspired models e.g., InceptionResNetV2 and ResNet18, benefit from residual/skip connections to improve the system performance and to prevent information loss among deeper designs. Herein, residual connections are considered to connect the output of a layer as the input of the following layers, for which an additional parameter or arrangement is not required. From a different perspective, the residual connectivity can be seen as a bypass operator among layers that ensures more efficient feature maps by preventing the increment in training error. In summary, ResNet-based architectures intend to remove the vanishing gradient problem, transfer the necessary information between transition points of layers, and produce more robust feature maps [23,27].

### 2.5. Dataset Information and Handicaps

The training part of the BraTS 2017/2018 library involves 210 HGG and 75 LGG samples defined in 3D MRI. In 3D data, there exist 3D images in the size of 240 × 240 × 155 per phase that is diversely presented for every phase (T1, T2, T1ce, FLAIR). In addition, 3D data include 155 slices of which every image is specified in 3D concerning the RGB space [14,15,16,23].

In the BraTS 2017/2018 training data, a tumor-based mask is defined as including four labels (background, non-enhancing and edema, peritumoral, and GD-enhancing) [14,15]. However, if the background and the other three labels are respectively assigned as ‘0’ and ‘1’, the 3D-defined tumor is found without sub-dividing the tumor into sub-regions. In other words, we evaluate the tumor as a unique section in our study, and considering the information in tissues, the tumor sub-regions are seen as tantamount to each other to preserve the necessary information [16,23]. Furthermore, this tumor extraction includes the logic of a segmentation section of a fully automated CAD, and 3t2FTS-v2 can be directly applicable to the extracted 3D data. In our paper, the classification part of a CAD is considered utilizing the 3t2FTS-v2 approach (Figure 1) to reveal the 2D-ID images and the ResNet50 model (Figure 2) to classify the 2D-ID information [23,27].

Figure 3 describes the disadvantages of the dataset used in terms of the 3D and cross-sectional perspectives [16,23]. As seen in the 3D-based representation,

A tumor type (LGG or HGG) can have very different size and shape features if the examination is performed inside one type. On the contrary, if HGG and LGG-type tumors are examined together, the shape-based and size-based features can be similar.

According to the cross-sectional-based determination,

A tumor type can have very different intensity features inside the tumor, which can be similar to the intensity features of the opposite tumor type.

In summary, there is no distinctive information that can be considered to categorize the tumor types. Regarding the disadvantages of data originating from the nature of glioma, a robust classification model should be proposed by examining the size, shape, intensity, and location features among all slices of 3D tumor data. Concerning this, the importance of 3t2FTS-v2 can be better deduced from the disadvantage analyses of gliomas.

## 3. Experimental Analyses and Interpretations

In this study, ResNet50 is fixed as the classifier unit to be utilized with 3t2FTS-v2 considering the results and advice given previously in [23]. In other words, ResNet50 arises as the preponderant method in [23] among seven transfer learning architectures (DenseNet201, InceptionResNetV2, InceptionV3, ResNet101, SqueezeNet, VGG19, and Xception) for using with 3t2FTS-v1 for 3D tumor analyses. Herein, this performance encourages the usage of ResNet50 to operate with 3t2FTS-v2.

Three strategies are examined to design the normalization part of 3t2FTS-v2, as declared in Figure 1, which are the options of *without normalization*, *minmax normalization*, and *z-score normalization*. Herein, the appropriate normalization option is analyzed to be coherent with the proposed feature transform strategy, and the key point is whether the normalization approach should be utilized, as it is not available in 3t2FTS-v1. In addition, 3t2FTS-v2 also considers GLRLM features [25]. Moreover, other extraction approaches (GLCM, etc.) have been examined for the formation of the 3t2FTS-v2 by yielding that other preferences are not appropriate to use with the 3D tumor information.

The 2D-ID images produced with 3t2FTS-v2 are presented as the input of ResNet50 architecture to discriminate the HGG/LGG labels. The hyperparameters of ResNet50 are comprehensively examined as shown in Table 1, according to previous studies [23,27,28], for observing the highest performance that can be achieved. Among hyperparameters, the epoch is fixed as 100 to prevent memory errors. All experiments are performed using the default values of other hyperparameters in the *Deep Network Designer toolbox* of the *MATLAB* software. In experiments, two-fold cross-validation is preferred to test the model performance [23]. Classification accuracy is chosen as the unique evaluation metric in terms of its significance to objectively evaluate the hyperparameter arrangements and to reveal the appropriate normalization preference.

Table 2, Table 3, and Table 4, respectively, present the ResNet50 results for three options (*without normalization*, *with minmax normalization*, and *with z-score normalization*) in 3t2FTS-v2.

As seen in Table 2, the highest accuracy (78.59%) is observed whilst the optimizer, LRDF, learning rate, and mini-batch size are, respectively, chosen as *rmsprop*, ‘0.8’, ‘0.0001’, and ‘32’. If an in-depth evaluation is performed regarding the average accuracies, it was seen that *adam* and *rmsprop* optimizers outperform *sgdm* preference by obtaining approximately 75% accuracy among 16 trials. Among 24 trials, the most appropriate preferences of the learning rate and mini-batch size are ‘0.001’ and ‘32’, respectively, achieving 75.01% and 75.27% accuracy scores, respectively. For LRDF preferences, the choice of ‘0.8’ seems more appropriate to use concerning the achieved average accuracy of 75.08% among 12 trials. In terms of the average accuracy-based examinations, ResNet50 reaches a 74.77% average accuracy among 48 trials on data formed *without normalization* preference for 3t2FTS-v2.

Regarding Table 3, the best accuracy (82.45%) is recorded when optimizer, LRDF, learning rate, and mini-batch size are, respectively, preferred as *rmsprop*, ‘0.6’, ‘0.0001’, and ‘32’. In the event of average accuracy-based evaluation, it was revealed that *adam* stays as the best optimizer by providing a 76.59% accuracy among 16 trials. Among 24 trials, the reliable choices of the learning rate and mini-batch size seem to be ‘0.0001’ and ‘32’ by attaining 76.28% and 75.13% accuracies, respectively. The LRDF choice of ‘0.2’ was better than the other preferences by scoring a 75.43% accuracy in 12 trials. By means of the average accuracy-based assessments, ResNet50 obtains a 75.09% average accuracy among 48 trials on data revealed with *minmax normalization* preference for 3t2FTS-v2.

Concerning Table 4, the highest accuracy score (99.64%) comes at three different arrangements for four hyperparameters. In these arrangements, the mini-batch size, learning rate, LRDF, and optimizer are, respectively, declared as ‘16’, ‘0.001’, ‘0.8’, *rmsprop* or ‘32’, ‘0.001’, ‘0.8’, *adam* or ‘32’, ‘0.0001’, ‘0.2’, *adam* for the highest performance. For average accuracy-based considerations, the *adam* optimizer is the best preference providing a 98.72% accuracy among 16 trials. Among 24 trials, the most appropriate preferences of the learning rate and mini-batch size are ‘0.001’ and ‘16’ by ensuring 97.62% and 97.70% accuracy scores, respectively. The preference of ‘0.2’ is the best LRDF value that achieves a 97.86% average accuracy. With regard to the average accuracy-based exploratory, ResNet50 acquires a 97.59% accuracy among 48 trials on data shaped with the *z-score normalization* choice for 3t2FTS-v2.

## 4. Discussions

In this study, the evaluation is performed on the basis of a two-fold cross-validation approach, and no data augmentation is considered, which is also not appropriate to use with the output of the 3t2FTS-v2 algorithm (2D-ID images). In other words, the proposed model is assessed with sufficient data, and the data imbalance is intended to be kept at its lowest level by using two-fold cross-validation. According to the performance assessments in Section 3, it was revealed that:Among average accuracy-based experiments, ResNet50 generally inclines to operate with the *adam* optimizer and LRDF value of ‘0.2’ (especially for normalization-available trials). Moreover, there is no discriminative adjustment for other hyperparameters in average performance-based analyses.Regarding the *z-score* normalization-based and average accuracy-based evaluations, the appropriate preferences of mini-batch size, learning rate, LRDF, and optimizer are, respectively, ‘16’, ‘0.001’, ‘0.2’, and *adam*.Concerning the highest scores observed, ResNet50 usually utilizes the mini-batch size of ‘32’, LRDF of ‘0.8’, and an *adam* or *rmsprop* optimizer.In relation to the *z-score* normalization-based and highest scores-based assessments, there is no discriminative information about the three adjustments. However, a mini-batch size of ‘32’, a learning rate of ‘0.001’, an LRDF of ‘0.8’, and *adam* optimizer are used twice for the obtainment of the highest scores.

Figure 4 presents the performance comparison and summarization of 3t2FTS-v2 options via the highest and average accuracy scores.

According to the performance evaluation in Figure 4, it was seen that:*Z-score* normalization reveals the most appropriate preference on the highest accuracy-based considerations by yielding 17.19% and 21.05% more accuracy than the *minmax* normalization and non-normalization choices, respectively.By means of average accuracy-based evaluations, the *z-score* normalization keeps its superiority by providing 22.50% and 22.82% more accuracy than the *minmax* normalization and non-normalization preferences, respectively.

*Z-score* normalization seems to be the most promising arrangement in 3t2FTS-v2, and also the 2D-ID images produced with this preference can own negative values inside. In other words, the matrix output of the *z-score* can include negative information values specified in identity images. Concerning the huge performance difference between *z-score* normalization and other choices, it can be inferred that the negative values in 2D-ID images let the ResNet50 model understand the identity of tumor type in a comprehensive manner. In brief, the proposed 3t2FTS-v2 emerges as a brilliant space transform strategy and as a new breakpoint to classify 3D tumors in MRI. 

With a different perspective to the analyses in 3D MRI, the proposed 3t2FTS-v2 algorithm is also suggested in the form of being utilized for the different types of tumors. Furthermore, 3t2FTS-v2 seems promising to use with the 3D brain-based classification to diagnose various diseases in brain voxels.

Table 5 shows the literature comparison of our model with the state-of-the-art systems considering HGG/LGG categorization. As shown in Table 5, only a few studies have evaluated the 3D-based categorization of HGG/LGG samples. Concerning Table 5, it was revealed that the proposed model operating the 3t2FTS-v2 approach and ResNet50 architecture outperforms all other systems, which are 2D-based, 2.5D-based, or 3D-based classification frameworks. In addition, the proposed method also achieves a 19.64% higher accuracy than that shown in a previous study [23], in which 3t2FTS-v1 is considered to generate 2D-ID inputs for ResNet50. At this point, the significance of FOS and GLRLM features and *z-score* normalization is proven to generate effective 2D-ID images in 3D voxel classification in 3D MRI data. Moreover, the proposed model achieves a remarkable performance on the imbalanced BraTS 2017/2018 dataset by proving its robustness via high classification performance. ResNet50 architecture proves its efficiency again to operate with 2D-ID-based identity images to categorize the glioma grades as in [23].

In summary, the usage of 3t2FTS-v2 and ResNet50 algorithms is presented in the literature, and the observed results will lead to various studies to analyze 3D voxel classification in 3D MRI data.

## 5. Conclusions

In this paper, a comprehensive survey is proposed about 3D tumor classification using 3D MRI data. By designing a remarkable space transform strategy (3t2FTS-v2), a new breakpoint is realized in 3D MRI-based voxel classification by enabling the usage of traditional machine learning algorithms that are on the basis of two-dimensional operations. In terms of evaluating the features in 3D and converting them to the 2D-based space, 3t2FTS-v2 yields a remarkable performance and constitutes the main novelty of our paper.

Regarding the experiments, ResNet50 seems as an appropriate transfer learning model to operate with the proposed 3t2FTS-v2 algorithm. In addition, a learning rate of ‘0.001’ and *adam* optimizer arise as the most frequent hyperparameter preferences in the model. The proposed model proves its superiority to the state-of-the-art studies by recording a 99.64% classification accuracy on the categorization of brain tumors defined in 3D space. Herein, task fulfillment with high performance is offered as another novelty of our paper. As mentioned, the proposed model approves oneself as being a fully automated classification section of a CAD system.

Concerning the nature of the 3t2FTS-v2 algorithm, the usage of the proposed method can be extended, and the following items enlighten the literature for future works:A comprehensive survey about 3t2FTS-v2 and its application for 3D brain-based disease categorization by using traditional machine learning algorithms or deep learning-based architecturesAn in-depth study utilizing 3t2FTS-v2 to classify various kinds of brain tumors on a large dataset by utilizing traditional machine learning algorithms or deep learning-based architecturesThe design of a novel deep learning architecture to operate with the 2D-ID identity images generated using 3t2FTS-v2

In addition to the aforementioned deductions, the usage of new datasets including noisy 3D images constitutes another research application. Furthermore, the data of patients with gamma-knife treatment can also be used to perform another qualified study, which both evaluates the longitudinal MRI data and the treatment planning [29,30].

## Figures and Tables

**Figure 1 bioengineering-10-00629-f001:**
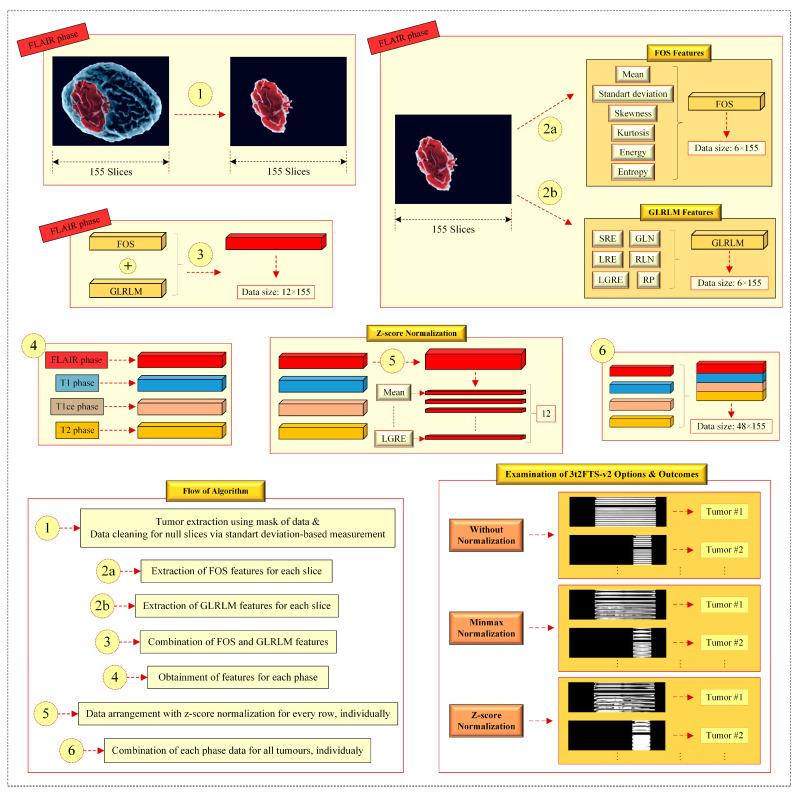
Design of the 3t2FTS-v2.

**Figure 2 bioengineering-10-00629-f002:**
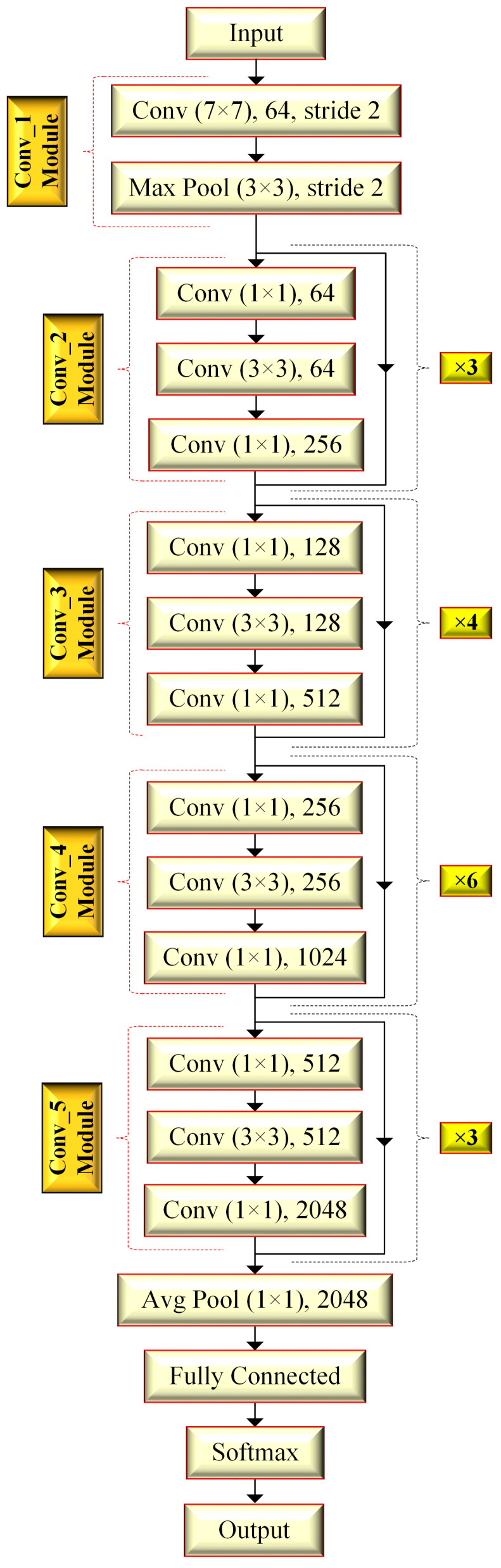
ResNet50 architecture [23,27].

**Figure 3 bioengineering-10-00629-f003:**
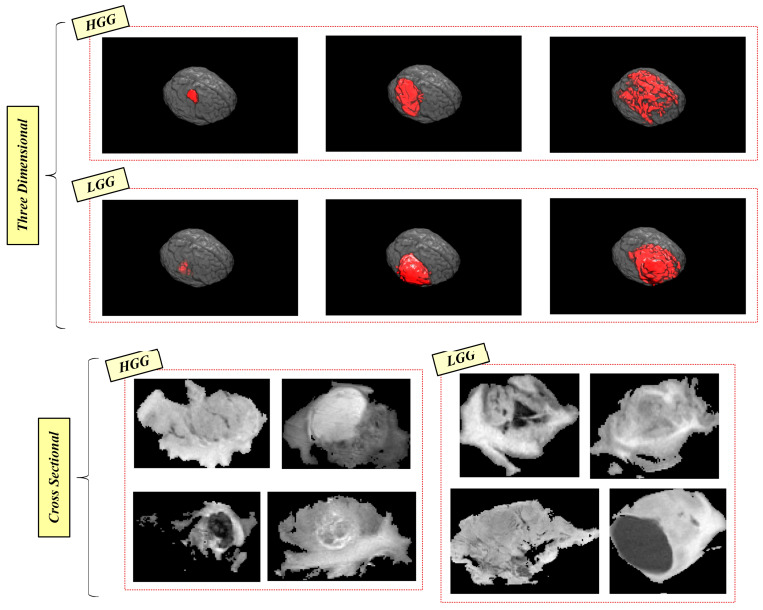
Visualization of disadvantages in the dataset [16,23].

**Figure 4 bioengineering-10-00629-f004:**
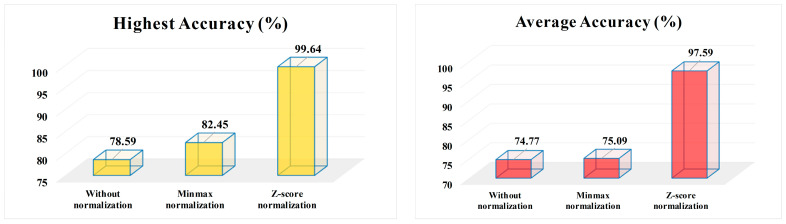
Summarization of 3t2FTS-v2 options via the highest and average accuracy scores.

**Table 1 bioengineering-10-00629-t001:** Hyperparameter arrangements of ResNet50.

Hyperparameter	Value/Preference
Epoch	100
Mini-batch size	16, 32
Learning rate	0.001, 0.0001
Learning Rate Drop Factor (LRDF)	0.2, 0.4, 0.6, 0.8
Optimizer	Adam, Rmsprop, Sgdm

**Table 2 bioengineering-10-00629-t002:** ResNet50 results without normalization in 3t2FTS-v2.

Mini-Batch Size	Learning Rate	LRDF	Optimizer	Accuracy	Learning Rate	LRDF	Optimizer	Accuracy
16	0.001	0.2	Adam	77.19	0.0001	0.2	Adam	74.73
			Sgdm	72.63			Sgdm	74.03
			Rmsprop	73.70			Rmsprop	73.33
		0.4	Adam	75.08		0.4	Adam	70.90
			Sgdm	77.19			Sgdm	71.92
			Rmsprop	74.73			Rmsprop	71.22
		0.6	Adam	73.68		0.6	Adam	76.14
			Sgdm	75.43			Sgdm	72.98
			Rmsprop	74.03			Rmsprop	72.98
		0.8	Adam	77.54		0.8	Adam	76.14
			Sgdm	71.92			Sgdm	75.78
			Rmsprop	76.14			Rmsprop	72.98
32	0.001	0.2	Adam	76.49	0.0001	0.2	Adam	74.73
			Sgdm	75.08			Sgdm	73.68
			Rmsprop	75.78			Rmsprop	77.19
		0.4	Adam	77.89		0.4	Adam	76.14
			Sgdm	73.68			Sgdm	74.03
			Rmsprop	75.78			Rmsprop	74.73
		0.6	Adam	70.87		0.6	Adam	76.14
			Sgdm	75.78			Sgdm	74.73
			Rmsprop	76.49			Rmsprop	76.84
		0.8	Adam	72.63		0.8	Adam	74.38
			Sgdm	74.03			Sgdm	74.38
			Rmsprop	76.49			Rmsprop	*78.59*

**Table 3 bioengineering-10-00629-t003:** ResNet50 results with minmax normalization in 3t2FTS-v2.

Mini-Batch Size	Learning Rate	LRDF	Optimizer	Accuracy	Learning Rate	LRDF	Optimizer	Accuracy
16	0.001	0.2	Adam	75.78	0.0001	0.2	Adam	77.54
			Sgdm	75.08			Sgdm	74.03
			Rmsprop	69.47			Rmsprop	75.43
		0.4	Adam	79.64		0.4	Adam	78.24
			Sgdm	76.49			Sgdm	72.98
			Rmsprop	64.56			Rmsprop	75.78
		0.6	Adam	76.14		0.6	Adam	76.14
			Sgdm	77.89			Sgdm	73.33
			Rmsprop	73.68			Rmsprop	76.49
		0.8	Adam	73.68		0.8	Adam	75.78
			Sgdm	77.54			Sgdm	75.43
			Rmsprop	73.68			Rmsprop	76.14
32	0.001	0.2	Adam	77.89	0.0001	0.2	Adam	79.29
			Sgdm	73.33			Sgdm	73.33
			Rmsprop	74.73			Rmsprop	79.29
		0.4	Adam	76.14		0.4	Adam	78.59
			Sgdm	78.59			Sgdm	72.98
			Rmsprop	71.92			Rmsprop	75.08
		0.6	Adam	74.57		0.6	Adam	75.08
			Sgdm	78.24			Sgdm	72.63
			Rmsprop	62.10			Rmsprop	*82.45*
		0.8	Adam	70.52		0.8	Adam	80.35
			Sgdm	75.43			Sgdm	72.63
			Rmsprop	66.31			Rmsprop	81.75

**Table 4 bioengineering-10-00629-t004:** ResNet50 results with z-score normalization in 3t2FTS-v2.

Mini-Batch Size	Learning Rate	LRDF	Optimizer	Accuracy	Learning Rate	LRDF	Optimizer	Accuracy
16	0.001	0.2	Adam	98.95	0.0001	0.2	Adam	98.24
			Sgdm	98.95			Sgdm	97.19
			Rmsprop	94.40			Rmsprop	98.59
		0.4	Adam	96.80		0.4	Adam	98.94
			Sgdm	98.59			Sgdm	96.14
			Rmsprop	95.40			Rmsprop	98.94
		0.6	Adam	98.59		0.6	Adam	99.29
			Sgdm	98.59			Sgdm	96.49
			Rmsprop	91.57			Rmsprop	99.29
		0.8	Adam	99.29		0.8	Adam	97.54
			Sgdm	98.95			Sgdm	95.08
			Rmsprop	*99.64*			Rmsprop	99.29
32	0.001	0.2	Adam	98.59	0.0001	0.2	Adam	*99.64*
			Sgdm	98.59			Sgdm	92.98
			Rmsprop	99.29			Rmsprop	98.94
		0.4	Adam	99.29		0.4	Adam	99.29
			Sgdm	98.24			Sgdm	91.92
			Rmsprop	98.59			Rmsprop	98.59
		0.6	Adam	97.54		0.6	Adam	98.94
			Sgdm	98.59			Sgdm	94.03
			Rmsprop	94.73			Rmsprop	98.94
		0.8	Adam	*99.64*		0.8	Adam	98.94
			Sgdm	98.59			Sgdm	95.08
			Rmsprop	91.57			Rmsprop	98.94

**Table 5 bioengineering-10-00629-t005:** Literature comparison.

Study	Year	Classification System	Dataset	Task	Accuracy (%)
Koyuncu et al. [16]	2020	The framework including three phase information (T1, T2, FLAIR), FOS, Wilcoxon ranking, and GM-CPSO-NN	210 HGG/75 LGG (BraTS 2017/2018)	3D-based classification (HGG vs. LGG)	90.18
Mzoughi et al. [17]	2020	A model operating 3D Deep CNN, data augmentation, and T1ce phase information	210 HGG/75 LGG (BraTS 2017/2018)	3D-based classification (HGG vs. LGG)	96.49
Tripathi and Bag [18]	2022	A model utilizing ResNets fusion with a novel DST and T2 phase information	2304 HGG/5088 LGG (TCIA)	2D-based classification (HGG vs. LGG)	95.87
Montaha et al. [19]	2022	A model using TD-CNN-LSTM and all phase information	234 HGG/74 LGG (BraTS 2020)	3D-based classification (HGG vs. LGG)	98.90
Jeong et al. [20]	2022	A model determining multimodal fusion network with adversarial learning and all phase information	210 HGG/75 LGG (BraTS 2017/2018)	2.5D-based classification (HGG vs. LGG)	90.91
Bhatele and Bhadauria [21]	2023	A model comprising DWT, GGLCM, LBP, GLRLM, morphological features, PCA, ensemble classifier, and all phase information	Not clearly defined (BraTS 2013)	2D-based classification (HGG vs. LGG)	100
			220 HGG/54 LGG (BraTS 2015)		99.52
Demir et al. [22]	2023	A model considering 3ACL and all phase information	220 HGG/54 LGG (BraTS 2015)	3D-based classification (HGG vs. LGG)	98.90
			210 HGG/75 LGG (BraTS 2017/2018)		99.29
Hajmohamad and Koyuncu [23]	2023	A model evaluating 3t2FTS and ResNet50	210 HGG/75 LGG (BraTS 2017/2018)	3D-based classification (HGG vs. LGG)	80
** *This study* **	2023	A model examining 3t2FTS-v2 and ResNet50	210 HGG/75 LGG (BraTS 2017/2018)	3D-based classification (HGG vs. LGG)	** *99.64* **

## Data Availability

In this study, we used a publicly available MRI dataset: BraTS 2017/2018.

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
