# Peer review of "A New Breakpoint to Classify 3D Voxels in MRI: A Space Transform Strategy with 3t2FTS-v2 and Its Application for ResNet50-Based Categorization of Brain Tumors"

_bioengineering, 2023, doi:10.3390/bioengineering10060629_

Round 1

Reviewer 1 Report

This paper presents a 3D voxel classification approach.
1. Figure 1 illustrates the design of the proposed 3t2FTS-v2. It is not clear what the new things are in this figure.
2. It is not recommended to use the abbreviation "3t2FTS" in the paper title.
3. In Section 2.1, it would be good to remove the symbol '' to avoid confusion in the mathematical notations.
4. Other than ResNet50, can the proposed method be applied to other baseline models?
5. Figure 2 illustrates the standard ResNet50 model; it can be removed.
6. Table 5 presents the literature results; it should be moved to the literature review part.
7. A few hand-crafted features are used in the proposed approach, such as FOS, GLRLM, etc. It would be good to conduct a feature importance study to figure out how these features influenced the final model recognition result.

 Minor editing of English language required

Author Response

Dear Editor,

Thank you very much for your contributions. 

Please see the file named "List_of_changes (for Reviewer#1).pdf".

Best regards and wish you a healthy day,

Reviewer 2 Report

Overall an interesting contribution, with a good quality of the presentation, and complete experimental trials. In order to enhance completeness and correctness, some concerns must be approached and corrected.

comment 1 (abstract)

The following sentence must be reworked in order to enhance its meaning.

"However, except for a new approach named 3t2FTS, a promising feature transform (from 3D to two-dimensional (2D) space) is not efficiently investigated for classification applications in 3D magnetic resonance imaging (3D MRI)."

comment 2 (1. Introduction)

The sentence "Also, magnetic resonance imaging (MRI) is the most frequently used modality in brain tumor-based segmentation or classification systems." doesn't provide a complete overview. MRI is used in brain imaging mainly to his ability to differentiate among different soft tissues.

comment 3 (2. Materials and Methods)

Why only FOS and GLRLM features have been used?

comment 4 (2. Materials and Methods)

In "As seen in the 3D-based representation, it’s seen that; 261

• A current tumour type (LGG or HGG) can have very different size and shape fea-262 tures. On contrary, if HGG and LGG type tumours are examined together, the shape and 263 size features can be similar to each other. 264

According to cross-sectional-based determination, it’s seen that; 265

• A tumor type can own very different intensity features inside of the tumor which 266 can be similar to the opposite tumor type."

I cannot understand the label in Figure 1. What do the authors intend with 'motivation'?

Moreover, the meaning of these sentences is not enough clear.

comment 5 (2.5. Dataset Information and Handicaps)

Why a bulleted list composed of only 1 item is used?

comment (3. Experimental Analyses and Interpretations)

To enhance the presentation's quality, I suggest using boldface numbers in Tables 2, 3, and 4 to highlight the best results.

comment 6 (5. Conclusions)

Besides the HGG/LGG classification, do the authors intend to apply the proposed approach in any other clinical context?

Patients to be treated with gamma-knife could represent an interesting field of application where there are three different areas to be classified (enhancement, necrosis, oedema) before treatment planning. The papers suggested below, properly discussed and referenced, could be useful for the authors to introduce this case study as a future application/development.

https://doi.org/10.1002/ima.22139

https://doi.org/10.1002/ima.22253

comment 7 (whole manuscript)

An overall English revision is mandatory in order to enhance language correctness and fluency. 

Author Response

Dear Reviewer#2,

Thank you very much for your contributions. 

Please see the file named "List_of_changes (for Reviewer#2).pdf".

Best regards and wish you a healthy day,

Round 2

Reviewer 1 Report

The revision is fine, there is no further comments.

NA

Author Response

Dear Reviewer#1,

We thank you very much for your relevance, helps, and contributions.

Best regards,

Reviewer 2 Report

Comment 6 was not addressed. I understand the answer of the authors, but to provide the reader with a view of the work done and possible future developments, it would be useful to integrate the conclusions appropriately. The literature works suggested represent interesting insights, to be discussed in the conclusion considering future developments, and can only improve the reader's perspective view of the study made.

Author Response

Dear Reviewer#2,

We thank you very much for your relevance, helps, and contributions.

Best regards,

Round 3

Reviewer 2 Report

All comments have been properly approached.